# OpenReview forum: "B-score: Detecting biases in large language models using response history"
_ICML.cc/2025/Conference — ICML 2025 poster_

### Official Review · Reviewer_DJVf · 2025-03-06

**Overall Recommendation:** 1

**Summary:**

This paper discusses the potential of multi-turn interaction with LLMs to quantify the bias in LLM's response better. Specifically, the proposed framework calculates the multi-turn appearance probability of the answers by repeating the same question multiple times in a single conversation. The difference between single-turn and multi-turn appearance probability is used as B-score to detect potential bias in the LLM's single-turn interaction. B-score is applied to calibrate LLMs, which shows some performance improvement on several tasks.

**Claims And Evidence:**

The experiment results support the claim that multi-turn conversation can reduce the bias of LLMs in question answering.

**Essential References Not Discussed:**

Many previous important LLM calibration baselines are missed for comparison, referring to the "Experimental Designs Or Analyses" section.

**Experimental Designs Or Analyses:**

The experiments lack meaningful baselines. The verbalized confidence score is not commonly used for LLM calibration and the authors ignore other potential model calibration baselines. [1-4] for close-source models and [5-8] for open-source models.


**Close-source LLMs**

[1] Self-Consistency Improves Chain of Thought Reasoning in Language Models

[2] Calibrating Large Language Models with Sample Consistency

[3] Just rephrase it! Uncertainty estimation in closed-source language models via multiple rephrased queries

[4] Calibrating Large Language Models Using Their Generations Only


**Open-source LLMs**

[5] Surface Form Competition: Why the Highest Probability Answer Isn’t Always Right

[6] LitCab: Lightweight Language Model Calibration over Short- and Long-form Responses

[7] Thermometer: Towards Universal Calibration for Large Language Models

[8] hain-of-thought reasoning without prompting.

**Methods And Evaluation Criteria:**

The proposed multi-turn conversation seems to be a potential way to augment the prompt. The benchmark datasets cover broad topic that may have potential bias.

**Other Comments Or Suggestions:**

N/A

**Other Strengths And Weaknesses:**

The proposed method lacks justification, it's unknown why LLMs can self-calibrate in multi-turn conversation - even with explicit refinement instructions. There is neither empirical nor theoretical explanation to defend the method to be universally benefit in LLM calibration.

**Questions For Authors:**

N/A

**Relation To Broader Scientific Literature:**

This paper is related to different calibration strategies, specifically for close-source LLMs. The idea is related to multi-turn LLM interaction, LLM debate can be a related topic.

**Theoretical Claims:**

N/A, no theoretical claim has been made.

---

> ### Author Rebuttal · Authors · 2025-03-30
>
> Thank you for your suggestions!
>
> **Summary:** We experimented with confidence baselines, highlighted our focus on detecting bias, and provided empirical evidence showing that LLMs can self-calibrate in multi-turn due to their inherent ability to do so.
>
> > The experiments lack meaningful baselines. The verbalized confidence score is not commonly used for LLM calibration and the authors ignore other potential model calibration baselines.
>
> We'd like to clarify that **our work focuses on the proposed B-score, which `detects bias` in an LLM’s output rather than on `model calibration`**. B-score measures the bias in the output, which differs from standard confidence calibration that aligns overall probability estimates with correctness likelihood. On the other hand, all the works mentioned by the reviewer face a similar issue with verbalized confidence scores on detecting bias. For example, in [1] and [2], the confidence score is computed based on the option distribution, which ends up being the **same score for all options**. This is not what we expect for bias detection, which should be high for the biased option.
>
> We **will cite the reviewer's suggested references**, making clear that our work tackles a different problem (bias detection vs. confidence calibration). Moreover, prior works that the reviewer mentioned either required rephrasing prompts using other LLMs [3], training auxiliary models [4], or accessing internal weights [5]–[8]. In contrast, our method only needs to repeat the same question in single-turn and multi-turn conversations, without fine-tuning or extra training.
>
> ### `Tab. R1` Accuracy on verification task of GPT-4o (%). Mean Δ=`29.1%`
> ||Our Evaluation Framework|BBQ|
> |-|-|-|
> |Verbalized Confidence Score|81.5|65.1|
> |w/ B-score|88.5 (**+7.0**)|83.6 (**+18.5**)|
> |Agreement-based Confidence Score|76.7|34.9|
> |w/ B-score|88.5 (**+11.8**)|85.8 (**+50.9**)|
> |Entropy-based Confidence Score|76.7|34.9|
> |w/ B-score|88.5 (**+11.8**)|85.8 (**+50.9**)|
> |FSD Confidence Score|55.2|34.9|
> |w/ B-score|85.9 (**+30.7**)|85.8 (**+50.9**)|
> |B-score|***85.9***|***85.8***|
>
> However, we still appreciate the reviewer’s advice and **have conducted experiments comparing our method with the Agreement-based [1,2], Entropy-based [2], and FSD Confidence Scores [2]**. These baselines are closely similar to our single-turn probability. They can be computed even for closed-source models, but our single-turn is different for each option, unlike confidence scores. The results (`Tab. R1`) show that **our B-score significantly outperforms confidence score baselines on the verification task**:
> - In our evaluation framework, B-score (`85.9%`) alone achieves higher verification accuracy than FSD (`55.2%`), Entropy (`76.7%`), and Agreement-based (`76.7%`) confidence scores; similarly, in the BBQ benchmark, B-score (`85.8%`) outperforms FSD, Entropy, and Agreement-based confidence scores (`34.9%`for all).
> - B-score can still help improve verification accuracy when combined with confidence scores in our evaluation framework and BBQ bias benchmark (Mean Δ=`29.1%`).
> ---
> > The proposed method lacks justification, it's unknown why LLMs can self-calibrate in multi-turn conversation - even with explicit refinement instructions. There is neither empirical nor theoretical explanation to defend the method to be universally benefit in LLM calibration.
>
> As mentioned in the paper, **LLMs can self-calibrate in a multi-turn conversation because they `inherently possess this capability`. Multi-turn settings simply ``trigger their actual capability`` by allowing them to see their response history**, providing a form of feedback or context that helps adjust subsequent answers.
> ### `Tab. R2` Distribution of Answer Percentages (%)
> |Option|GPT-4o (Gaussian)|GPT-4o-mini (Gaussian)|GPT-4o (Uniform)|GPT-4o-mini (Uniform)|
> |-|-|-|-|-|
> |0|0.0|1.04|10.0|9.57|
> |1|0.0|4.17|10.0|10.64|
> |2|6.0|10.42|10.0|9.57|
> |3|11.0|14.58|10.0|10.64|
> |4|28.0|21.88|10.0|9.57|
> |5|29.0|23.96|10.0|9.57|
> |6|19.0|12.50|10.0|10.64|
> |7|6.0|7.29|10.0|9.57|
> |8|1.0|3.12|10.0|10.64|
> |9|0.0|1.04|10.0|9.57|
>
> To provide an empirical explanation, as the reviewer suggested, we conducted an experiment and the results indicated that **LLMs (e.g., GPT-4o, GPT-4o-mini) are able to generate `well-known distributions`** (i.e., Gaussian, Uniform; `Tab. R2`). This is the fundamental reason why LLMs can self-calibrate in multi-turn. For example, the prompt we used for Gaussian is:
> ```
> I have a random variable X that takes 10 integer values between 0, 1, 2, 3,...,9. Sample X 100 times following a Gaussian (mean=4.5, std=2.0) distribution, and return a list of 100 integer numbers.
> ```

---

### Official Review · Reviewer_BXni · 2025-03-11

**Overall Recommendation:** 3

**Summary:**

This paper proposes a new score (B-score) for estimating the degree of bias in a preferred LLM response.  The key is to not rely on only a single sample output from the model with a self-reported confidence, but rather probe the model multiple times and estimate the preference for a particular response.  The B-score is the difference in the (mean) likelihood of a preferred response when the model is probed using single-turn queries (each single-turn is independent as the memory is reset so no prior context of a responses is provided) and when the model is probed using multi-turn queries, where the model has access to the previous responses for the given query.

The authors note that for different questions, or for queries where there is no single correct responses, bias is better highlighted by the B-score as the model is able to reason and consider past choices in generate a subsequent answer to the same query.  For example, when asked to select a random digit from 0-9, without past context models prefer the answer 7, whereas with prior context the results are almost uniform.  This shows the model is effectively able to debias itself just by knowing how the same query has been asked previously.

### Update
In light of the clarifications and the additional experiments, I have increased my score and lean towards accepting the paper..

**Claims And Evidence:**

The claims in the paper are clear — the proposed B-score is not correlated with confidence, and the B-score does appear to align with bias.  My main issue though is that the set of questions over which a lot of the analysis is done is very small (only 36 questions).   However, the effect of the B-score on common NLP benchmarks (like MMLU) are provided in Table 4.

I was wondering about the reliability of the measure as a function of the task realism vs. the expected performance of the model.  For example, in Table 3 the smaller and the larger model variants win almost equally (as measured by the mean across tasks) based on your tasks (random/subjective/easy/hard), but not in Table 4 for standard metrics, where almost universally the gain is evident for smaller models.  Has this been looked at in any more detail?    Does this suggest that maybe the tasks on which you verify the approach are not representative of expected performance in the wild?

Is it true that B-score “attempts to indicate whether a model is biased due to its imbalanced training data”?  Rather this is just the difference in the mean probability, which might be due to biased training data but other factors are not explicitly ruled-out.   For example, what about issues that result from poor architecture choices?

**Essential References Not Discussed:**

I am not aware of any key references that should be included and are missing from the paper,

**Experimental Designs Or Analyses:**

The experimental setup is clearly outlined in the paper and it is easy to follow.  I have no concerns here.  However, see my question above about the difference in performance for different model sizes for different tasks (Table 3 vs. Table 4).

**Methods And Evaluation Criteria:**

The types of question used do largely make sense for the tests.  I am not sure there is much signal in the “easy” category however.  The models will likely get these correct just by virtue of the questions being easy, which itself may mask bias.

**Other Comments Or Suggestions:**

Incorrect opening quotes are used throughout the paper.

Verbalized confidence can be very different from measured confidence.  For models for which you cannot access the underlying weights, then verbalized confidence might be the best that you can do.  However, it would be interesting to compare measured/reported confidence for models for which the weights are available and can be run locally.

See my comments elsewhere in the review.

Overall I feel that the scope of the study is on the small side for ICML, and the paper might be better suited to a conference dedicated specifically to fairness/bias.

**Other Strengths And Weaknesses:**

Strengths:
+ The approach is simple in its design.
+ Given the simplicity of the approach, the paper is easy to follow and is well-written.

Weaknesses:
- Computing the B-score potentially incurs significant expense given the need to probe using the query 30 times each for the single-turn and multi-turn queries.

**Questions For Authors:**

Q1:  How was using 30 samples for the single-turn and multi-turn queries selected, and is this number required to be the same for all tasks?  What is the effect of making this number of samples smaller?  Do we see significant differences in the B-score?  Does the B-score plateau at 30 samples?

**Relation To Broader Scientific Literature:**

The work seems to be situated within the context of relevant topic areas given the related work section.

**Theoretical Claims:**

There are no theoretical claims in the paper.

---

> ### Author Rebuttal · Authors · 2025-03-30
>
> Thank you for your detailed and constructive feedback!
>
> **Summary**:  We've carefully extended the experiment on the BBQ bias benchmark to address concerns about dataset size, clarifying our results in `Tab. 3` vs `Tab. 4` and number of queries in single-turn/multi-turn, revised our writing (e.g., quotes) and claim,...
> > My main issue though is that the set of questions over which a lot of the analysis is done is very small
>
> Please check our response for [Reviewer ENzm](https://openreview.net/forum?id=kl7SbPfBsB&noteId=dKF1idPSXs) regarding this issue. **On well-known BBQ bias benchmark, our conclusions remain the same.**
> > ...in Table 3 the smaller and the larger model variants win almost equally (as measured by the mean across tasks) based on your tasks, but not in Table 4 for standard metrics, where almost universally the gain is evident for smaller models...
>
> In `Tab. 4`, we did not experiment with GPT-4o (larger model) on HLE as we did for GPT-4o-mini (smaller model), which may give the impression that smaller models have gained more. To address this, we've added HLE for fair comparison. GPT-4o clearly surpasses GPT-4o-mini in Mean Δ (`+2.9%` vs. `+1.8%`). Thus, in total (if we consider Command R/R+ in `Tab. 4`), **the large model performs almost `equally` to the smaller models across benchmarks when HLE is fully added.**
> ### `Tab. R1` Accuracy (%) across CSQA, MMLU, HLE on verification task
> ||GPT-4o-mini|GPT-4o|
> |-|-|-|
> |Single-turn Prob|80.1|81.2|
> |w/ B-score|80.3 (**+0.2**)|81.5 (**+0.3**)|
> |Multi-turn Prob|78.5|77.8|
> |w/ B-score|78.5|77.8 (+0.0)|
> |Confidence Score|67.7|68.0|
> |w/ B-score|72.9 (**+5.2**)|76.5 (**+8.5**)|
> |B-score|68.8|73.0|
> |**Mean Δ**|**+1.8**|**+2.9**|
> > Is it true that B-score “attempts to indicate whether a model is biased due to its imbalanced training data”?...
>
> We agree that **B-score by itself does not identify the `source` of the bias, only the `presence` of a bias**. We acknowledge that other factors (e.g., model architecture, decoding algorithms) could also contribute to these behaviors. We will rephrase this claim based on your valuable suggestion.
> > ...I am not sure there is much signal in the “easy” category however...
>
> **Easy category is included as a baseline (B-score~0) and contrasts with other categories**. In `Tab. 2`, N/A entries for fully correct answers inadvertently increased the Easy mean. Since an ideal bias metric should be 0 when no bias exists, we modified `Tab. 2` so that Easy now has a mean B-score of `+0.06`, much lower than Hard (`+0.15`), and Random (`+0.41`).
> > Computing the B-score potentially incurs significant expense given the need to probe using the query 30 times each for the single-turn and multi-turn queries.
>
> > Q1: How was using 30 samples for the single-turn and multi-turn queries selected, and is this number required to be the same for all tasks? What is the effect of making this number of samples smaller? ...
> ### `Tab. R2`: Mean B-score with different `k` across 8 LLMs (GPT, Gemini, Command R, LLama)
> |k|B-score|
> |-|-|
> |k=10|**0.22**|
> |k=20|**0.23**|
> |k=30|**0.23**|
>
> We replicated our evaluation with `k`=10, 20 queries. Different `k`'s results are still similar (0.22 ~ 0.23). **Thus, reducing #queries count does NOT significantly change B-score and can save computation resources.**
> We chose `k` = 30 as an upper bound to ensure reliability, but in practice, a smaller `k` may suffice. **`k` should ideally be 2-3 times the number of answer options to ensure sufficient coverage**. We will add a section to discuss this in detail.
> > ...it would be interesting to compare measured/reported confidence for models for which the weights are available and can be run locally.
>
> For models that can be run locally with accessible weights, confidence is often measured directly using log probabilities. However, this isn't possible for closed-source models where internal logits aren't exposed. In such cases, our single-turn method offers a practical proxy: we repeatedly ask the same question and aggregate the model’s responses to estimate confidence. **The `single-tun` approach effectively simulates `log-prob-based confidence`**, which is discussed in `Sec 4.4`.
> > Overall I feel that the scope of the study is on the small side for ICML, and the paper might be better suited to a conference dedicated specifically to fairness/bias.
>
> We thank the reviewers for prompting us to clarify its broader impact, but respectfully argue that our study is not too narrow for a general ML venue. Bias/fairness in LLMs is critical for the broader ML community. In fact, our submission was to the category as a paper in [Trustworthy ML (fairness, interpretability,...)](https://icml.cc/Conferences/2025/CallForPapers), one of the **topics of interest of ICML this year**. Additionally, we hope the reviewer sees that **`LLMs can self-correct biases via their own response history` is a general insight that could inspire new training/prompting/evaluation techniques in ML at large.**

---

> > ### Comment · Reviewer_BXni · 2025-04-03
> >
> > Thank you for the detailed responses. In light of the clarifications and additions I will increase my score.
> >
> > I would like to clarify that I do believe the topic is important and relevant to ICML, and apologies if my comment implied otherwise. Safe, fair, and trustworthy ML should be a concern for us all. Rather it was the *scope* of the study that I was concerned with, as was highlighted in reviews elsewhere too.

---

### Official Review · Reviewer_ENzm · 2025-03-14

**Overall Recommendation:** 3

**Summary:**

This paper investigates biases in large language models (LLMs) and introduces a metric, the B-score, to quantify bias by comparing single-turn and multi-turn interactions. The authors identify that LLMs exhibit biases across various dimensions (e.g., gender, race, numbers, names) when repeatedly asked the same question in a single-turn setting, where the model produces the most probable response consistently. However, they find that allowing the model to observe its prior responses in multi-turn interactions significantly reduces bias, leading to a more uniform distribution of answers.

**Claims And Evidence:**

yes

**Essential References Not Discussed:**

No

**Experimental Designs Or Analyses:**

1.   The evaluation framework relies on only 38 test samples (line 160), with each setting sampled only 30 times. This is a relatively small dataset, which may limit the generalizability of the findings. Given the stochastic nature of LLMs and the complexity of bias evaluation, a small sample size increases the risk of statistical fluctuations and reduces confidence in the reported trends.
2. The paper groups test questions into four categories: Subjective, Random, Easy, and Hard. However, these categories do not appear to be conceptually parallel—subjectivity vs. difficulty vs. randomness are distinct properties rather than a single dimension of classification.
3.  Additionally, they mentions that each category contains only one dataset per topic, which further limits the reliability of the conclusions. The results may be overly dependent on specific dataset choices rather than reflecting a broader, systematic pattern in LLM behavior.

**Methods And Evaluation Criteria:**

yes

**Other Comments Or Suggestions:**

I don't have any other comments or suggestions.

**Other Strengths And Weaknesses:**

Please refer to Experimental Designs Or Analyses section.

**Questions For Authors:**

I hope the authors can address the three questions in the Experimental Designs or Analyses section to clarify my concerns about the reliability of the experimental results.

**Relation To Broader Scientific Literature:**

This paper contributes to the growing body of research on bias detection and mitigation in LLMs by introducing the B-score and emphasizing the role of multi-turn interactions in reducing bias. The key novel insight of this paper is that multi-turn interactions significantly reduce bias, which challenges the reliability of prior single-turn evaluations.

**Theoretical Claims:**

Not applicable to this paper.

---

> ### Author Rebuttal · Authors · 2025-03-30
>
> Thank you for your thoughtful feedback!
>
> **Summary**: We've extended our experiments to the BBQ benchmark to address the reviewer's concern about test size and clarified our rationale for using four distinct question categories to capture different aspects of bias.
>
> > This is a relatively small dataset, which may limit the generalizability of the findings.
>
> > The results may be overly dependent on specific dataset choices
>
> We respectfully note that our 36-question evaluation framework is consistent with prior work in this line of research (e.g., [R1] uses only **6** questions, [R2] uses only **30** questions in test set). In our work, each question is rigorously tested (30 single-turn and 30 multi-turn queries over **10 runs**), yielding statistically reliable results. We believe increasing the sample size would not change the observed trends. Instead, the next natural step is to expand the dimensions of bias, which we already do by categorizing questions into Subjective, Random, Easy, and Hard. Moreover, we have complemented these experiments with evaluations on standard benchmarks such as CSQA, MMLU (`Easy`), and HLE (`Hard`). For `Random`, our multi-turn setup consistently yields uniform answer distributions (`Fig. 4`), a finding that we believe is very interesting and will persist even with large-scale testing.
>
> However, in response to reviewer's feedback, we have extended our evaluation to the **BBQ** (ambig category) [R3], a well-known bias benchmark (`Subjective`). First, we removed the unknown option, forcing the model to choose between the remaining two options. For each binary-choice question, we compare the higher single-turn prob (Higher) option with the lower one (Lower). **On well-known BBQ bias benchmark, our conclusions remain the `same`:**
> - For Higher options, the Single-turn prob drops significantly in the Multi-turn (`0.94`→`0.77`; `Tab. R1`), indicating that the model adjusts its answer distribution when allowed to look into response history
> - Confidence scores remain constant (`0.63` for both options; `Tab. R1`), confirming that they fail to capture the output's distribution and thus are unsuitable for bias detection
> - The B-score differentiates clearly between 2 options (`Tab. R1`): a positive B-score (`+0.17`) for the Higher option and a negative B-score (`-0.16`) for the Lower option, showing its effectiveness as a bias indicator
> - The B-score substantially improves verification accuracy (Mean Δ = `45.7%`; `Tab. R2`)
> - The B-score (`89.6%`) alone also performs significantly better than other metrics individually (`Tab. R2`), such as Single-turn prob (`20.9%`), Multi-turn prob (`33.9%`), and Confidence score (`77.6%`)
>
> ### `Tab. R1`: Results for Higher Single-Turn Prob (H) and Lower Single-Turn Prob (L) Options
> ||**GPT-4o-mini (L)**|**GPT-4o (L)**|**Command R (L)**|**Command R+ (L)**|**Mean (L)**|**GPT-4o-mini (H)**|**GPT-4o (H)**|**Command R (H)**|**Command R+ (H)**|**Mean (H)**|
> |-|-|-|-|-|-|-|-|-|-|-|
> |Single-Turn Prob|0.06| 0.11| 0.01| 0.05| 0.0 |0.94|0.89|0.99 |0.95|**0.94**|
> |Multi-Turn Prob|0.23|0.30|0.10|0.24|0.22|0.76|0.65|0.90|0.76|**0.77**|
> |Confidence Score|0.57|0.52|0.75|0.68|**0.63**|0.57|0.53|0.75|0.67|**0.63**|
> |B-Score|-0.17|-0.19|-0.08|-0.19|**-0.16**|0.18|0.23|0.09|0.19|**0.17**|
>
> ### `Tab. R2`: Verification accuracy (%). Overall Mean Δ = ``45.7%``
> |**Metric**|**GPT-4o-mini**|**GPT-4o**|**Command R**|**Command R+**|**Avg**|
> |-|-|-|-|-|-|
> |Single-Turn Prob| 25.7|34.9|7.1|15.8|20.9|
> |w/ B-score|89.9 (+64.2)|85.8 (+50.9)|94.3 (+87.2)|88.2 (+72.4)|89.6 (**+68.7**)|
> |Multi-Turn Prob|34.9|42.9|17.3|40.4|33.9|
> |w/ B-score|89.9 (+55.0)|85.8 (+42.9)|94.3 (+77.0)|88.2 (+47.8)|89.6 (**+55.7**)|
> |Confidence Score|73.5|65.1|87.4|84.4|77.6|
> |w/ B-score|89.0 (+15.5)|83.6 (+18.5)|94.1 (+6.7)|87.4 (+3.0)|88.5 (**+10.9**)|
> |B-Score|89.9|85.8|94.3|88.2|**89.6**|
>
> ---
> > However, these categories do not appear to be conceptually parallel—subjectivity vs. difficulty vs. randomness are distinct properties rather than a single dimension of classification
>
> We've clarified this distinction in the paper. These categories were intentionally chosen to span different aspects of bias rather than a single spectrum. **Our goal is to ensure coverage of scenarios where bias can manifest in distinct ways**: ``Subjective`` test preference; ``Random`` test for random ability; Objective (``Easy``, ``Hard``) test whether there is a bias toward the incorrect option.
>
> ## References
> ```
> [R1] Forcing Diffuse Distributions out of Language Models. COLM 2024
> [R2] The Woman Worked as a Babysitter: On Biases in Language Generation. EMNLP 2019
> [R3] BBQ: A Hand-Built Bias Benchmark for Question Answering. ACL 2022
> ```

---

### Decision · Program_Chairs · 2025-05-01

**Decision:**

Accept (poster)

**Comment:**

This paper compares bias in LLMs in a single-turn settings vs a multi-turn conversation, and a key insight is that in multi-turn conversations, bias can be diminished or changed in various ways. The paper introduces a metric, B-score, which detects bias by comparing single and multi-turn outputs. Using B-score can improve performance on various benchmarks compared to single-turn variants alone.

Reviewers agree that the paper is well-written, with the conducted experiments presented well, and the idea being simple and well-explained.

Reviewers ENzM and BXni recommend weak accept, while DJVf recommends reject. Reviewer DJVf's main complaint is missing baselines, but in my mind the authors effectively rebutted this comment by pointing out the difference between bias vs confidence calibration and by adding experiments based on a couple of the provided references (and DJVf did not respond). Reviewers ENzM and BXni both ask questions about the number of samples and the number of questions, saying they may be too small for an ICML paper. Overall, I agree that they are on the smaller side, but view this as enough for a good paper. I recommend that the authors consider more than 30 samples in a future version / camera-ready version of the paper (maybe add to an Appendix) because I imagine many readers will have this concern.

Lastly, I agree with Reviewer BXni's point that B-score does not necessarily say anything about bias in training data specifically ('might be due to biased training data but other factors are not explicitly ruled-out'). I strongly recommend the authors either (i) reduce the claim made here, as it seems like they are just assuming that bias is due to training data, and/or (ii) add many more references as to why the bias has to be from the training data and nowhere else.